# Generalized Random Utility Models with Multiple Types

**Hossein Azari Soufiani**
SEAS
Harvard University
azari@fas.harvard.edu

**Hansheng Diao**
Mathematics Department
Harvard University
diao@fas.harvard.edu

**Zhenyu Lai**
Economics Department
Harvard University
zlai@fas.harvard.edu

**David C. Parkes**
SEAS
Harvard University
parkes@eecs.harvard.edu

## Abstract

We propose a model for demand estimation in multi-agent, differentiated product settings and present an estimation algorithm that uses reversible jump MCMC techniques to classify agents' types. Our model extends the popular setup in Berry, Levinsohn and Pakes (1995) to allow for the data-driven classification of agents' types using agent-level data. We focus on applications involving data on agents' ranking over alternatives, and present theoretical conditions that establish the identifiability of the model and uni-modality of the likelihood/posterior. Results on both real and simulated data provide support for the scalability of our approach.

## 1   Introduction

Random utility models (RUM), which presume agent utility to be composed of a deterministic component and a stochastic unobserved error component, are frequently used to model choices by individuals over alternatives. In this paper, we focus on applications where the data is rankings by individuals over alternatives. Examples from economics include the popular random coefficients logit model [7] where the data may involve a (partial) consumer ranking of products [9]. In a RUM, each agent receives an intrinsic utility that is common across all agents for a given choice of alternative, a pairwise-specific utility that varies with the interaction between agent characteristics and the characteristics of the agent's chosen alternative, as well as an agent-specific taste shock (noise) for his chosen alternative. These ingredients are used to construct a posterior/likelihood function of specific data moments, such as the fraction of agents of each type that choose each alternative.

To estimate preferences across heterogenous agents, one approach as allowed by prior work [20, 24] is to assume a mixture of agents with a finite number of types. We build upon this work by developing an algorithm to endogenously learn the classification of agent types within this mixture. Empirical researchers are increasingly being presented with rich data on the choices made by individuals, and asked to classify these agents into different types [28, 29] and to estimate the preferences of each type [10, 23]. Examples of individual-level data used in economics include household purchases from supermarket-scanner data [1, 21], and patients' hospital or treatment choices from healthcare data [22].

The partitioning of agents into latent, discrete sets (or "types") allows for the study of the underlying distribution of preferences across a population of heterogeneous agents. For example, preferences may be correlated with an agent characteristic, such as income, and the true classification of each agent's type, such as his income bracket, may be unobserved. By using a model of demand to estimate the elasticity in behavioral response of each type of agent and by aggregating these responses over the different types of agents, it is possible to simulate the impact of a social or public policy [8], or simulate the counterfactual outcome of changing the options available to agents [19].

## 1.1 Our Contributions

This paper focuses on estimating generalized random utility models (GRUM[1]) when the observed data is partial orders of agents' rankings over alternatives and when latent types are present.

We build on recent work [3, 4] on estimating GRUMs by allowing for an interaction between agent characteristics and the characteristics of the agent's chosen alternative.The interaction term helps us to avoid unrealistic substitution patterns due to the independence of irrelevant alternatives [26] by allowing agent utilities to be correlated across alternatives with similar characteristics. For example, this prevents a situation where removing the top choices of both a rich household and a poor household lead them to become equally likely to substitute to the same alternative choice. Our model also allows the marginal utilities associated with the characteristics of alternatives to vary across agent types.

To classify agents' types and estimate the parameters associated with each type, we propose an algorithm involving a novel application of reversible jump Markov Chain Monte Carlo (RJMCMC) techniques. RJMCMC can be used for model selection and learning a posterior on the number of types in a mixture model [31]. Here, we use RJMCMC to cluster agents into different types, where each type exhibits demand for alternatives based on different preferences; i.e., different interaction terms between agent and alternative characteristics.

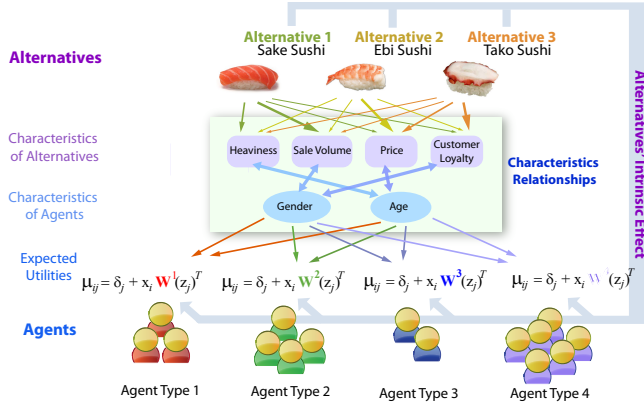

Figure 1: A GRUM with multiple types of agents

We apply the approach to a real-world dataset involving consumers' preference rankings and also conduct experiments on synthetic data to perform coverage analysis of RJMCMC. The results show that our method is scalable, and that the clustering of types provides a better fit to real world data. The proposed learning algorithm is based on Bayesian methods to find posteriors on the parameters. This differentiates us from previous estimation approaches in econometrics rely on techniques based on the generalized method of moments.[2]

The main theoretical contribution establishes identifiability of mixture models over data consisting of partial orders. Previous theoretical results have established identifiability for data consisting of vectors of real numbers [2, 18], but not for data consisting of partial orders. We establish conditions under which the GRUM likelihood function is uni-modal for the case of observable types. We do not provide results on the log concavity of the general likelihood problem with unknown types and leave it for future studies.

## 1.2 Related work

Prior work in econometrics has focused on developing models that use data aggregated across types of agents, such as at the level of a geographic market, and that allow heterogeneity by using random coefficients on either agents' preference parameters [7, 9] or on a set of dummy variables that define types of agents [6, 27], or by imposing additional structure on the covariance matrix of idiosyncratic taste shocks [16]. In practice, this approach typically relies on restrictive functional assumptions about the distribution of consumer taste shocks that enter the RUM in order to reduce computational

burden. For example, the logit model [26] assumes i.i.d. draws from a Type I extreme value distribution. This may lead to biased estimates, in particular when the number of alternatives grow large [5].

Previous work on clustering ranking data for variations of the Placket-Luce (PL) model [28, 29] has been restricted to settings without agent and alternative characteristics. Morover, Gormley et al. [28] and Chu et al. [14] performed clustering for RUMs with normal distributions, but this was limited to pairwise comparisons. Inference of GRUMs for partial ranks involved the computational hardness addressed in [3]. In mixture models, assuming an arbitrary number of types can lead to biased results, and reduces the statistical efficiency of the estimators [15].

To the best of our knowledge, we are the first to study the identifiability and inference of GRUMs with multiple types. Inference for GRUMs has been generalized in [4], However, Azari et al. [4] do not consider existence of multiple types. Our method applies to data involving individual-level observations, and partial orders with more than two alternatives. The inference method establishes a posterior on the number of types, resolving the common issue of how the researcher should select the number of types.

## 2 Model

Suppose we have $N$ agents and $M$ alternatives $\{c_1, .., c_M\}$, and there are $S$ types (subgroups) of agents and $s(n)$ is agent $n$'s type.

Agent characteristics are observed and defined as an $N \times K$ matrix $X$, and alternative characteristics are observed and defined as an $L \times M$ matrix $Z$, where $K$ and $L$ are the number of agent and alternative characteristics respectively.

Let $u_{nm}$ be agent $n$'s *perceived utility* for alternative $m$, and let $W^{s(n)}$ be a $K \times L$ real matrix that models the linear relation between the attributes of alternatives and the attributes of agents. We have,

$$u_{nm} = \delta_m + \vec{x}_n W^{s(n)} (\vec{z}_m)^T + \epsilon_{nm}, \tag{1}$$

where $\vec{x}_n$ is the $n$th row of the matrix $X$ and $\vec{z}_m$ is the $m$th column of the matrix $Z$. In words, agent $n$'s utility for alternative $m$ consists of the following three parts:

1. $\delta_m$:gs The *intrinsic utility* of alternative $m$, which is the same across all agents;

2. $\vec{x}_n W^{s(n)}(\vec{z}_m)^T$: The *agent-specific utility*, which is unique to all agents of type $s(n)$, and where $W^{s(n)}$ has at least one nonzero element;

3. $\epsilon_{nm}$: The *random noise* (agent-specific taste shock), which is generated independently across agents and alternatives.

The number of parameters for each type is $P = KL + M$.

See Figure 2 for an illustration of the model. In order to write the model as a linear regression, we define matrix $A^{(n)}_{M \times P}$, such that $A^{(n)}_{KL+m,m} = 1$ for $1 \leq m \leq M$ and $A^{(n)}_{KL+m,m'} = 0$ for $m \neq m'$ and $A^{(n)}_{(k-1)L+l,m} = \vec{x}_n(k)\vec{z}_m(l)$ for $1 \leq l \leq L$ and $1 \leq k \leq K$. We also need to shuffle the parameters for all types into a $P \times S$ matrix $\Psi$, such that $\Psi_{KL+m,s} = \delta$ and $\Psi_{(k-1)L+l,s} = W^s_{kl}$ for $1 \leq k \leq K$ and $1 \leq l \leq L$. We adopt $B^{(n)}_{S \times 1}$ to indicate the type of agent $n$, with $B^{(n)}_{s(n),1} = 1$ and $B^{(n)}_{s,1} = 0$ for all $s \neq s(n)$. We also define an $M \times 1$ matrix, $U^{(n)}$, as $U^{(n)}_{m,1} = u_{nm}$. We can now rewrite (1) as:

$$U^{(n)} = A^{(n)} \Psi B^{(n)} + \epsilon \tag{2}$$

Suppose that an agent has type $s$ with probability $\gamma_s$. Given this, the random utility model can be written as, $\Pr(U^{(n)}|X^{(n)}, Z, \Psi, \Gamma) = \sum_{s=1}^{S} \gamma_s \Pr(U^{(n)}|X^{(n)}, Z, \Psi^s)$, where $\Psi^s$ is the $s$th column of the matrix $\Psi$. An agent ranks the alternatives according to her perceived utilities for the alternatives. Define rank order $\pi^n$ as a permutation $(\pi^n(1), \ldots, \pi^n(m))$ of $\{1, \ldots, M\}$. $\pi^n$ represents the full ranking $[c_{\pi^i(1)} \succ_i c_{\pi^i(2)} \succ_i \cdots \succ_i c_{\pi^i(m)}]$ of the alternatives $\{c_1, .., c_M\}$. That is, for agent $n$, $c_{m_1} \succ_n c_{m_2}$ if and only if $u_{nm_1} > u_{nm_2}$ (In this model, situations with tied perceived utilities have zero probability measure).

The model for observed data $\pi^{(n)}$, can be written as:

$$\Pr(\pi^{(n)}|X^{(n)}, Z, \Gamma, \Psi) = \int_{\pi^{(n)}=order(U^{(n)})} \Pr(U^{(n)}|X^{(n)}, Z, \Psi, \Gamma) = \sum_{s=1}^{S} \gamma_s \Pr(\pi^{(n)}|X^{(n)}, Z, \Psi^s)$$

Note that $X^{(n)}$ and $Z$ are observed characteristics, while $\Gamma$ and $\Psi$ are unknown parameters. $\pi = order(U)$ is the ranking implied by U, and $\pi(i)$ is the $i$th largest utility in U. $D = \{\pi^1, .., \pi^N\}$ denotes the collection of all data for different agents. We have that

$$\Pr(D|X, Z, \Psi, \Gamma) = \prod_{n=1}^{N} \Pr(\pi^{(n)}|X^{(n)}, Z, \Psi, \Gamma)$$

## 3 Strict Log-concavity and Identifiability

In this section, we establish conditions for identifiability of the types and parameters for the model. Identifiability is a necessary property in order for researchers to be able to infer economically-relevant parameters from an econometric model. Establishing identifiability in a model with multiple types and ranking data requires a different approach from classical identifiability results for mixture models [2, 18, e.g.].

Moreover, we establish conditions for uni-modality of the likelihood for the parameters $\Gamma$ and $\Psi$, when the types are observed. Although our main focus is on data with unobservable types, establishing the conditions for uni-modality conditioned on known types remains an essential step because of the sampling and optimization aspects of RJMCMC. We sample from the parameters conditional on the algorithm's specification of types.

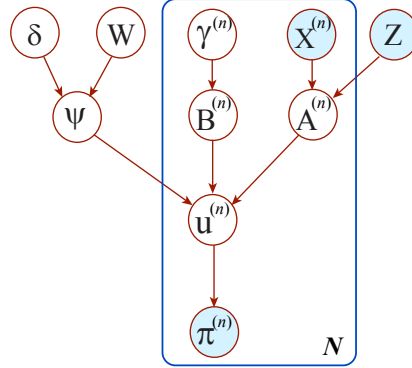

The uni-modality result establishes that the sampling approach is exploring a uni-modal distribution conditional on its specified types. Despite adopting a Bayesian point of view in presenting the model, we adopt a uniform prior on the parameter set, and only impose nontrivial priors on the number of types in order to obtain some regularization. Given this, we present the theory with regards to the likelihood function from the data rather than the posterior on parameters.

Figure 2: Graphical representation of the multiple type GRUM generative process.

### 3.1 Strict Log-concavity of the Likelihood Function

For agent $n$, we define a set $G^n$ of function $g^n$'s whose positivity is equivalent to giving an order $\pi^n$. More precisely, we define $g_m^n(\vec{\psi}, \vec{\epsilon}) = [\mu_{n\pi^n(m)} + \epsilon_{n\pi^n(m)}] - [\mu_{n\pi^n(m+1)} + \epsilon_{n\pi^n(m+1)}]$ for $m = 1, .., M - 1$ where $\mu_{nj} = \delta_j + \sum_{k,l} x_n(k) W_{kl}^{s(n)} z_j(l)$ for $1 \leq j \leq M$. Here, $\vec{\psi}$ is a vector of $KL + M$ variables consisting of all $\delta_j$'s and $W_{kl}$'s. We have, $L(\vec{\psi}, \pi^n) = L(\vec{\psi}, G^n) = \Pr(g_1^n(\vec{\psi}, \vec{\epsilon}) \geq 0, ..., g_{M-1}^n(\vec{\psi}, \vec{\epsilon}) \geq 0)$. This is because $g_m^n(\vec{\psi}, \vec{\epsilon}) \geq 0$ is equivalent to saying alternative $\pi^n(m)$ is preferred to alternative $\pi^n(m+1)$ in the RUM sense.

Then using the result in [3] and [30], $L(\vec{\psi}) = L(\vec{\psi}, \pi)$ is **logarithmic concave** in the sense that $L(\lambda\vec{\psi} + (1-\lambda)\vec{\psi}') \geq L(\psi)^{\lambda} L(\psi')^{1-\lambda}$ for any $0 < \lambda < 1$ and any two vectors $\vec{\psi}, \vec{\psi}' \in \mathbb{R}^{LK+M}$. The detailed statement and proof of this result are contained in the Appendix. Let's consider all $n$ agents together. We study the function, $l(\Psi, D) = \sum_{n=1}^{N} \log Pr(\pi^n|\vec{\psi}^{s(n)})$. By log-concavity of $L(\vec{\psi}, \pi)$ and using the fact that sum of concave functions is concave, we know that $l(\Psi, D)$ is concave in $\Psi$, viewed as a vector in $\mathbb{R}^{SKL+M}$. To show uni-modality, we need to prove that this

concave function has a unique maximum. Namely, we need to be able to establish the conditions for when the equality holds. If our data is subject to some mild condition, which implies boundedness of the parameter set that maximizes $l(\Psi, D)$, Theorem 1 bellow tells us when the equality holds. This condition has been explained in [3] as condition (1).

Before stating the main result, we define the following auxiliary $(M-1)N' \times (SKL + M - 1)$ matrix $\widetilde{A} = \widetilde{A}^{N'}$ (Here, let $N' \leq N$ be a positive number that we will specify later.) such that, $\widetilde{A}_{(M-1)(n-1)+m,(s-1)KL+(K-1)l+k}$ is equal to $x_n(k)(z_m(l) - z_M(l))$ if $s = s(n)$ and is equal to 0 if $s \neq s(n)$, for all $1 \leq n \leq N'$, $1 \leq m \leq M-1$, $1 \leq s \leq S$, $1 \leq k \leq K$, and $1 \leq l \leq L$. Also, $\widetilde{A}_{(M-1)(n-1)+m,SKL+m'}$ is equal to 1 if $m = m'$ and is equal to 0 if $m \neq m'$, for all $1 \leq m, m' \leq M-1$ and $1 \leq n \leq N'$.

**Theorem 1.** *Suppose there is an $N' \leq N$ such that rank $\widetilde{A}^{N'} = SKL + M - 1$. Then $l(\Psi) = l(\Psi, D)$ is **strictly concave up to $\delta$-shift**, in the sense that,*

$$l(\lambda \Psi + (1-\lambda)\Psi') \geq \lambda l(\Psi) + (1-\lambda)l(\Psi'), \tag{3}$$

*for any $0 < \lambda < 1$ and any $\Psi, \Psi' \in \mathbb{R}^{SKL+M}$, and the equality holds if and only if there exists $c \in \mathbb{R}$, such that:*

$$\begin{cases} \delta_m = \delta'_m + c & \text{for all } 1 \leq m \leq M \\ W^s_{kl} = W'^s_{kl} & \text{for all } s, k, l \end{cases}$$

The proof of this theorem is in the appendix.

**Remark 1.** *We remark that the strictness "up to $\delta$-shift" is natural. A $\delta$-shift results in a shift in the intrinsic utilities of all the products, which does not change the utility difference between products. So such a shift does not affect our outcome. In practice, we may set one of the $\delta$'s to be 0 and then our algorithm will converge to a single maximum.*

**Remark 2.** *It's easy to see that $N'$ must be larger than or equal to $1 + \frac{SKL}{M-1}$. The reason we introduce $N'$ is to avoid cumbersome calculations involving $N$.*

## 3.2 Identifiability of the Model

In this section, we show that, for the case of unobserved types, our model is identifiable for a certain class of cdfs for the noise in random utility models. Let's first specify this class of "nice" cdfs:

**Definition 1.** *Let $\phi(x)$ be a smooth pdf defined on $\mathbb{R}$ or $[0, \infty)$, and let $\Phi(x)$ be the associated cdf. For each $i \geq 1$, we write $\phi^{(i)}(x)$ for the $i$-th derivative of $\phi(x)$. Let $g_i(x) = \frac{\phi^{(i+1)}(x)}{\phi^{(i)}(x)}$. The function $\Phi$ is called **nice** if it satisfies one of the following two mutually exclusive conditions:*

(a) *$\phi(x)$ is defined on $\mathbb{R}$. For any $x_1, x_2 \in \mathbb{R}$, the sequence $\frac{g_i(x_1)}{g_i(x_2)}$ converges to some value in $\mathbb{R}$ (as $i \to \infty$) only if either $x_1 = x_2$; or $x_1 = -x_2$ and $\frac{g_i(x_1)}{g_i(x_2)} \to -1$ as $i \to \infty$.*

(b) *$\phi(x)$ is defined on $[0, \infty)$. For any $x_1, x_2 \geq 0$, the ratio $\frac{\phi^{(i)}(x_1)}{\phi^{(i)}(x_2)}$ is independent of $i$ for $i$ sufficiently large. Moreover, we require that $\phi(x_1) = \phi(x_2)$ if and only if $x_1 = x_2$.*

This class of nice functions contains normal distributions and exponential distributions. A proof of this fact is included in the appendix.

Identifiability is formalized as follows: Let $\mathcal{C} = \{\{\gamma_s\}_{s=1}^S \mid S \in \mathbb{Z}_{>0}, \gamma_i \in \mathbb{R}_{>0}, \sum_{s=1}^S \gamma_s = 1\}$. Suppose, for two sequences $\{\gamma_s\}_{s=1}^S$ and $\{\gamma'_s\}_{s=1}^{S'}$, we have:

$$\sum_{s=1}^S \gamma_s \Pr(\pi | X^{(n)}, Z, \Psi) = \sum_{s=1}^{S'} \gamma'_s \Pr(\pi | X^{(n)}, Z, \Psi') \tag{4}$$

for all possible orders $\pi$ of $M$ products, and for all agents $n$. Then, we must have $S = S'$ and (up to a permutation of indices $\{1, \cdots, S\}$) $\gamma_s = \gamma'_s$ and $\Psi = \Psi'$ (up to $\delta$-shift).

For now, let's fix the number of agent characteristics, $K$. One observation is that the number $x_n(k)$, for any characteristic $k$, reflects certain characteristics of agent $n$. Varying the agent $n$, this amount $x_n(k)$ is in a bounded interval in $\mathbb{R}$. Suppose the collection of data $D$ is sufficiently large. Based on this, assuming that $N$ can be be arbitrarily large, we can assume that the $x_n(k)$'s form a dense subset in a closed interval $I_k \subset \mathbb{R}$. Hence, (4) should hold for any $X \in I_k$, leading to the following theorem:

**Theorem 2.** *Define an* $(M-1) \times L$ *matrix* $\widetilde{Z}$ *by setting* $\widetilde{Z}_{m,l} = z_m(l) - z_M(l)$. *Suppose the matrix* $\widetilde{Z}$ *has rank L, and suppose,*

$$\sum_{s=1}^{S} \gamma_s \Pr(\pi|X, Z, \Psi) = \sum_{s=1}^{S'} \gamma_s' \Pr(\pi|X, Z, \Psi'), \tag{5}$$

*for all* $x(k) \in I_k$ *and all possible orders* $\pi$ *of M products. Here, the probability measure is associated with a nice cdf. Then we must have* $S = S'$ *and (up to a permutation of indices* $\{1, \cdots, S\}$*),* $\gamma_s = \gamma_s'$ *and* $\Psi = \Psi'$ *(up to* $\delta$*-shift).*

The proof of this theorem is provided in the appendix. Here, we illustrate the idea for the simple case, with two alternatives ($m = 2$) and no agent or alternative characteristics ($K = L = 1$). Equation (5) is merely a single identity. Unwrapping the definition, we obtain:

$$\sum_{s=1}^{S} \gamma_s \Pr(\epsilon_1 - \epsilon_2 > \delta_1 - \delta_2 + xW^s(z_1 - z_2)) = \sum_{s=1}^{S'} \gamma_s' \Pr(\epsilon_1 - \epsilon_2 > \delta_1' - \delta_2' + xW'^s(z_1 - z_2)). \tag{6}$$

Without loss of generality, we may assume $z_1 = 1$, $z_2 = 0$, and $\delta_2 = 0$. We may further assume that the interval $I = I_1$ contains 0. (Otherwise, we just need to shift $I$ and $\delta$ accordingly.) Given this, the problem reduces to the following lemma:

**Lemma 1.** *Let* $\Phi(x)$ *be a nice cdf. Suppose,*

$$\sum_{s=1}^{S} \gamma_s \Phi(\delta + xW^s) = \sum_{s=1}^{S'} \gamma_s' \Phi(\delta' + xW'^s), \tag{7}$$

*for all* $x$ *in a closed interval* $I$ *containing* 0. *Then we must have* $S = S'$, $\delta = \delta'$ *and (up to a permutation of* $\{1, \cdots, S\}$*)* $\gamma_s = \gamma_s$, $W^s = W'^s$.

The proof of this lemma is in the appendix. By applying this to (6), we can show identifiablity for the simple case of $m = 2$ and $K = L = 1$.

Theorem 2 guarantees identifiability in the limit case that we observe agents with characteristics that are dense in an interval. Beyond the theoretical guarantee, we would in practice expect (6) to have a unique solution with a enough agents with different characteristics. Lemma 1 itself is a new identifiability result for scalar observations from a set of truncated distributions.

## 4 RJMCMC for Parameter Estimation

We are using a uniform prior for the parameter space and regularize the number of types with a geometric prior. We use a Gibbs sampler, as detailed in the appendix (supplementary material Algorithm (1)) to sample from the posterior. In each of $T$ iterations, we sample utilities $u^n$ for each agent, matrix $\psi_s$ for each type, and set of assignments of agents to alternatives $\mathbf{S}^n$. The utility of each agent for each alternative conditioned on the data and other parameters is sampled from a truncated Exponential Family (e.g. Normal) distribution. In order to sample agent $i$'s utility for alternative $j$ ($u_{ij}$), we set thresholds for lower and upper truncation based on agent $i$'s former samples of utility for the two alternatives that are ranked one below and one above alternative $j$, respectively.

We use reversible-jump MCMC [17] for sampling from conditional distributions of the assignment function (see Algorithm 1). We consider three possible moves for sampling from the assignment function $\mathbf{S}(n)$:

(1) Increasing the number of types by one, through moving a random agent to a new type of its own. The acceptance ratio for this move is: $\Pr_{split} = \min\{1, \frac{\Pr(S+1)\Pr(\mathcal{M}^{(t+1)}|D)}{\Pr(S)\Pr(\mathcal{M}^{(t)}|D)} \cdot \frac{\frac{1}{S+1}}{\frac{1}{S}} \cdot \frac{p_{+1}}{p_{-1}} \cdot \frac{1}{p(\alpha)} \cdot \mathcal{J}_{(t)\to(t+1)}\}$, where $\mathcal{M}^{(t)} = \{u, \psi, B, \mathbf{S}, \pi\}^{(t)}$, and $\mathcal{J}_{(t)\to(t+1)} = 2^P$ is the Jacobian of the transformation from the previous state to the proposed state and $\Pr(S)$ is the prior (regularizer) for the number of types.

(2) Decrease the number of types by one, through merging two random types. The acceptance ratio for the merge move is: $\Pr_{merge} = \min\{1, \frac{\Pr(S-1)\Pr(\mathcal{M}^{(t+1)}|D)}{\Pr(S)\Pr(\mathcal{M}^{(t)}|D)} \cdot \frac{\frac{1}{S-1}}{\frac{1}{S}} \cdot \frac{p_{-1}}{p_{+1}} \cdot \mathcal{J}_{(t)\to(t+1)}\}$.

(3) We do not change the number of types, and consider moving one random agent from one type to another. This case reduces to a standard Metropolis-Hastings, where because of the normal symmetric proposal distribution, the proposal is accepted with probability: $\Pr_{mh} = \min\{1, \frac{\Pr(\mathcal{M}^{(t+1)}|D)}{\Pr(\mathcal{M}^{(t)}|D)}\}$.

## 5  Experimental Study

We evaluate the performance of the algorithm on synthetic data, and for a real world data set in which we observe agents' characteristics and their orderings on alternatives. For the synthetic data, we generate data with different numbers of types and perform RJMCMC in order to estimate the parameters and number of types. The algorithm is implemented in MATLAB and scales linearly in the number of samples and agents. It takes on average $60 \pm 5$ seconds to generate 50 samples for $N = 200$, $M = 10$, $K = 4$ and $L = 3$ on an i5 2.70GHz Intel(R).

**Coverage Analysis for the number of types $S$ for Synthetic Data:** In this experiment, the data is generated from a randomly chosen number of clusters $S$ for $N = 200$, $K = 3$, $L = 3$ and $M = 10$ and the posterior on $S$ is estimated using RJMCMC. The prior is chosen to be $\Pr(S) \propto \exp(-3SKL)$. We consider a noisy regime by generating data from noise level of $\sigma = 1$, where all the characteristics $(X, Z)$ are generated from $\mathcal{N}(0, 1)$. We repeat the experiment 100 times. Given this, we estimate 60%, 90% and 95% confidence intervals for the number of types from the posterior samples. We also estimate the *coverage* percentage,

---

**Algorithm 1** RJMCMC to update $\mathbf{S}^{(t+1)}(n)$ from $\mathbf{S}^{(t)}(n)$

---

Set $p_{-1}, p_0, p_{+1}$, Find $S$: number of distinct types in $\mathbf{S}^{(t)}(n)$
Propose move $\nu$ from $\{-1, 0, +1\}$ with probabilities $p_{-1}, p_0, p_{+1}$, respectively.
**case $\nu = +1$:**
  Select random type $M_s$ and agent $n \in M_s$ uniformly and Assign $n$ to module $M_{s_1}$ and remainder to $M_{s_2}$ and Draw vector $\alpha \sim \mathcal{N}(0, 1)$ and Propose $\psi_{s_1} = \psi_s - \alpha$ and $\psi_{s_2} = \psi_s + \alpha$ and Compute proposal $\{u^n, \pi^n\}^{(t+1)}$
  Accept $\mathbf{S}^{(t+1)}(M_{s_1}) = S + 1$, $\mathbf{S}^{(t+1)}(M_{s_2}) = s$ with $\Pr_{split}$ from update $S = S + 1$
**case $\nu = -1$:**
  Select two random types $M_{s_1}$ and $M_{s_2}$ and Merge into one type $M_s$ and Propose $\psi_s = (\psi_{s_1} + \psi_{s_1})/2$ and Compute proposed $\{u^n, \pi^n\}^{(i+1)}$
  Accept $\mathbf{S}^{(t+1)}(n) = s_1$ for $\forall n$ s.t. $\mathbf{S}^{(t)}(n) = s_2$ with $\Pr_{merge}$ update $S = S - 1$
**case $\nu = 0$:**
  Select two random types $M_{s_1}$ and $M_{s_2}$ and Move a random agent $n$ from $M_{s_1}$ to $M_{s_2}$ and Compute proposed $\{u^{(n)}, \pi^{(n)}\}^{(t+1)}$
  Accept $\mathbf{S}^{(t+1)}(n) = s_2$ with probability $\Pr_{mh}$
**end switch**

---

which is defined to be the percentage of samples which include the true number of types in the interval. The simulations show 61%, 73%, 88%, 93% for the intervals 60%, 75%, 90%, 95% respectively, which indicates that the method is providing reliable intervals for the number of types.

**Performance for Synthetic Data:** We generate data randomly from a model with between 1 and 4 types. $N$ is set to 200, and $M$ is set to 10 for $K = 4$ and $L = 3$. We draw $10,000$ samples from the stationary posterior distribution. The prior for $S$ has chosen to be $\exp(-\alpha SKL)$ where $\alpha$ is uniformly chosen in $(0, 10)$. We repeat the experiment 5 times. Table 1 shows that the algorithm successfully provides larger log posterior when the number of types is the number of true types.

**Clustering Performance for Real World Data:** We have tested our algorithm on a sushi dataset, where $5,000$ users provide rankings on $M = 10$ different kinds of sushi [25]. We fit the multi-type

GRUM for different number of types, on 100 randomly chosen subsets of the sushi data with size $N = 200$ , using the same prior we used in synthetic case and provide the performance on the Sushi data in Table 1. It can be seen that GRUM with 3 types has significantly better performance in terms of log posterior (with the prior that we chose, log posterior can be seen as log likelihood penalized for number of parameters) than GRUM with one, two or four types. We have taken non-categorical features as $K = 4$ feature for agents (age, time for filling the questionnaire, region ID, prefecture ID) and $L = 3$ features for sushi ( price,heaviness, sales volume).

## 6 Conclusions

In this paper, we have proposed an extension of GRUMs in which we allow agents to adopt heterogeneous types. We develop a theory establishing the identifiability of the mixture model when we observe ranking data. Our theoretical results for identifiability show that the number of types and the parameters associated with them can be identified. Moreover, we prove uni-modality of the likelihood (or posterior) function when types are observable. We propose a scalable algorithm for inference, which can be parallelized for use on very large data sets. Our experimental results show that models with multiple types provide a significantly better fit, in real-world data. By clustering agents into multiple types, our estimation algorithm allows choices to be correlated across agents of the same type, without making any *a priori* assumptions on how types of agents are to be partitioned. This use of machine learning tech-

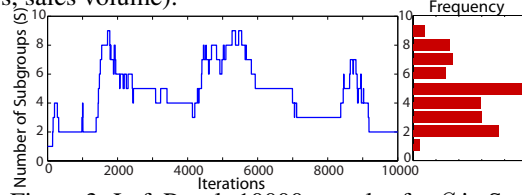

Figure 3: Left Panel: 10000 samples for $S$ in Synthetic data, where the true S is $5$. Right Panel: Histogram of the samples for $S$ with max at $5$ and mean at $4.56$.

| Type | Synthetic True types | | | | Sushi |
| --- | --- | --- | --- | --- | --- |
| | One | two | Three | Four | sushi |
| one type | **-2069** | -2631 | -2780 | -2907 | -2880 |
| two types | -2755 | **-2522** | **-2545** | -2692 | -2849 |
| three types | -2796 | -2642 | **-2582** | -2790 | **-2819** |
| four types | -2778 | -2807 | -2803 | **-2593** | -2850 |

Table 1: Performance of the method for different number of true types and number of types in algorithm in terms of log posterior. All the standard deviations are between $15$ and $20$. Bold numbers indicate the best performance in their column with statistical significance of 95%.

niques complements various approaches in economics [11, 7, 8] by allowing the researcher to have additional flexibility in dealing with missing data or unobserved agent characteristics. We expect the development of these techniques to grow in importance as large, individual-level datasets become increasingly available. In future research we intend to pursue applications of this method to problems of economic interest.

## Acknowledgments

This work is supported in part by NSF Grants No. CCF- 0915016 and No. AF-1301976. We thank Elham Azizi for helping in the design and implementation of RJMCMC algorithm. We thank Simon Lunagomez for helpful discussion on RJMCMC. We thank Lirong Xia, Gregory Lewis, Edoardo Airoldi, Ryan Adams and Nikhil Agarwal for comments on the modeling and algorithmic aspects of this paper. We thank anonymous NIPS-13 reviewers, for helpful comments and suggestions.

## Footnotes

[1]Defined in [4] as a RUM with a generalized linear model for the regression of the mean parameters on the interaction of characteristics data as in Figure 1

[2]There are alternative methods to RJMCMC, such as the saturation method [13]. However, the memory required to keep track of former sampled memberships in the saturation method quickly becomes infeasible given the combinatorial nature of our problem.

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
