[Supplementary Material]

# Supplementary Material for: Generalized Random Utility Models with Multiple Types

**Hossein Azari Soufiani**
SEAS
Harvard University
azari@fas.harvard.edu

**Hansheng Diao**
Mathematics Department
Harvard University
diao@fas.harvard.edu

**Zhenyu Lai**
Economics Department
Harvard University
zlai@fas.harvard.edu

**David C. Parkes**
SEAS
Harvard University
parkes@eecs.harvard.edu

## 1 Theory: Strict Log-concavity and Identifiability

Despite the presentation of the model and inference from a Baysian point of view, we adopt a uniform prior on the parameter set and only impose nontrivial priors on the number of types, in order to obtain some regularization. Given this, we present the theory in regard to the likelihood function on data rather than the posterior on parameters.

The following section explores the conditions required to guarantee uni-modality of the likelihood and identifiability of the parameter space. We note here that the likelihood function is not uni-modal for the general model with unobserved agent types that was proposed in the former section. This is due to an extreme flexibility in combinatorial choice of types and parameters. However, given the observed types, we can show that the likelihood function is uni-modal for the parameters $\Psi$. In the first part of this section, we establish the uni-modality of the likelihood function.

The model and algorithm in this paper considers types unknown and learns the number of types. Establishing the conditions for uni-modality conditioned on known types is essential. This is due to the sampling/optimization aspect of the algorithm. In our algorithms, we sample from the parameters conditional on the algorithm's specification of types. Our results establishes that our sampling algorithm is exploring a uni-modal distribution conditional on its specified types.

Moreover, we establish conditions for identifiability of the model. Identifiability is a necessary property in order for the researcher to be able to infer economically-relevant parameters from any econometric model. Establishing identifiability on the model with multiple types on ranking data requires a different approach from classical identifiability results for mixture models [1, 3, e.g.], because we observe discrete data (ranking data) along with characteristics of agents rather than scalars or vector of scalars. We establish the identifiability of the parameters $\Gamma$ and $\Psi$, and number of types for the general model for the case of a finite number of types.

### 1.1 Strict Log-concavity of the Likelihood Function

First, let's fix an agent $n \in \{1, ..., N\}$. Define a set $G^n$ of function $g^n$'s whose positivity is equivalent to giving an order $\pi^n$. More precisely, define $g_m^n(\vec{\psi}, \vec{\epsilon}) = [\mu_{n\pi^n(m)} + \epsilon_{n\pi^n(m)}] - [\mu_{n\pi^n(m+1)} + \epsilon_{n\pi^n(m+1)}]$ for $m = 1, .., M-1$ where $\mu_{nj} = \delta_j + \sum_{k,l} x_n(k) W_{kl}^{s(n)} z_j(l)$ for $1 \le j \le M$. Here $\vec{\psi}$ is a vector of $KL + M$ variables consisting of all $\delta_j$'s and $W_{kl}$'s. Clearly, the length of each order

$\pi^n$ is $M - 1$. We have:

$$L(\vec{\psi}, \pi^n) = L(\vec{\psi}, G^n) = \Pr(g_1^n(\vec{\psi}, \vec{\epsilon}) \geq 0, ..., g_{M-1}^n(\vec{\psi}, \vec{\epsilon}) \geq 0) \tag{1}$$

This is because $g_m^n(\vec{\psi}, \vec{\epsilon}) \geq 0$ is equivalent to saying alternative $\pi^n(m)$ is preferred to alternative $\pi^n(m+1)$ in the RUM sense. Under this setting, we have the following generalization of the main result in [2]. Since we are now dealing with a single $n$, we drop the upper index $n$ and $s(n)$.

**Lemma 1.** *Suppose $\vec{\epsilon}$ is a vector of $M$ real numbers that are generated according to a distribution whose pdf is strictly logarithmic concave in $\mathbb{R}^M$. Consider the function*

$$L(\vec{\psi}, \pi) = L(\vec{\psi}, G) = \Pr(g_1(\vec{\psi}, \vec{\epsilon}) \geq 0, ..., g_{M-1}(\vec{\psi}, \vec{\epsilon}) \geq 0) \tag{2}$$

*Then $L(\vec{\psi}) = L(\vec{\psi}, \pi)$ is **logarithmic concave** in the sense that $L(\lambda\vec{\psi} + (1 - \lambda)\vec{\psi'}) \geq L(\psi)^\lambda L(\psi')^{1-\lambda}$ for any $0 < \lambda < 1$ and any two vectors $\vec{\psi}, \vec{\psi'} \in \mathbb{R}^{LK+M}$.*

The proof of this lemma is in section 2.

Now let's consider all $n$ agents together. We study the function, $l(\Psi, D) = \sum_{n=1}^N \log L(\vec{\psi}^{s(n)}, \pi^n) = \sum_{n=1}^N \log Pr(\pi^n | \vec{\psi}^{s(n)})$. By Lemma 1 and using the fact that sum of concave functions is concave, we know that $l(\Psi, D)$ is concave in $\Psi$, viewed as a vector in $\mathbb{R}^{SP}$. To show uni-modality, we need to prove that this concave function has a unique maximum. Namely, we need to be able to describe when the equality holds in the previous lemma. Actually, if our data is subject to some mild condition, which implies boundedness of the parameter set that maximizes $l(\Psi, D)$, Theorem 1 bellow tells us exactly when the equality holds. This condition has been explained in [2] as condition (1).

Before stating the main result, we define the following auxiliary $(M - 1)N' \times (SKL + M - 1)$ matrix $\widetilde{A} = \widetilde{A}^{N'}$ (Here $N' \leq N$ is a positive number we are going to specify later) such that, $\widetilde{A}_{(M-1)(n-1)+m,(s-1)KL+(K-1)l+k}$ is equal to $x_n(k)(z_m(l) - z_M(l))$ if $s = s(n)$ and is equal to 0 if $s \neq s(n)$, for all $1 \leq n \leq N'$, $1 \leq m \leq M - 1$, $1 \leq s \leq S$, $1 \leq k \leq K$, and $1 \leq l \leq L$. And, $\widetilde{A}_{(M-1)(n-1)+m,SKL+m'}$ is equal to 1 if $m = m'$ and is equal to 0 if $m \neq m'$, for all $1 \leq m, m' \leq M - 1$ and $1 \leq n \leq N'$.

**Theorem 1.** *Suppose there is an $N' \leq N$ such that rank $\widetilde{A}^{N'} = SKL + M - 1$. Then $l(\Psi) = l(\Psi, D)$ is **strictly concave up to $\delta$-shift**, in the sense that,*

$$l(\lambda\Psi + (1 - \lambda)\Psi') \geq \lambda l(\Psi) + (1 - \lambda)l(\Psi'), \tag{3}$$

*for any $0 < \lambda < 1$ and any $\Psi, \Psi' \in \mathbb{R}^{SP}$, and the equality holds if and only if there exists $c \in \mathbb{R}$, such that:*

$$\begin{cases} \delta_m = \delta'_m + c & \text{for all } 1 \leq m \leq M \\ W_{kl}^s = W_{kl}'^s & \text{for all } s, k, l \end{cases}$$

The proof of this theorem is in section 2.

**Remark 1.** *We remark that the strictness "up to $\delta$-shift" is natural. A $\delta$-shift results in a shift in the intrinsic utilities of all the products, which does not change the utility difference between products. So such a shift does not affect our outcome. In practice, we may set one of the $\delta$'s to be $0$ and then the simulation will converge to a single maximum.*

**Remark 2.** *It's easy to see that $N'$ must be larger than or equal to $1 + \frac{SKL}{M-1}$. The reason we introduce $N'$ is to avoid cumbersome calculation involving $N$.*

## 1.2 Identifiability of the Model

In practice, it is often the case that we do not know the number of types, and do not observe agent types. In this section, we show that, under this situation, our model is identifiable for a certain class of cdf. Let's first specify this class of "nice" cdfs:

**Definition 1.** *Let $\phi(x)$ be a smooth pdf defined on $\mathbb{R}$ or $[0, \infty)$, and let $\Phi(x)$ be the associated cdf. For each $i \geq 1$, we write $\phi^{(i)}(x)$ for the $i$-th derivative of $\phi(x)$. Let $g_i(x) = \frac{\phi^{(i+1)}(x)}{\phi^{(i)}(x)}$. The function $\Phi$ is called **nice** if it satisfies one of the following two mutually exclusive conditions:*

(a) $\phi(x)$ is defined on $\mathbb{R}$. For any $x_1, x_2 \in \mathbb{R}$, the sequence $\frac{g_i(x_1)}{g_i(x_2)}$ converges to some value in $\mathbb{R}$ (as $i \to \infty$) only if either $x_1 = x_2$; or $x_1 = -x_2$ and $\frac{g_i(x_1)}{g_i(x_2)} \to -1$ as $i \to \infty$.

(b) $\phi(x)$ is defined on $[0, \infty)$. For any $x_1, x_2 \geq 0$, the ratio $\frac{\phi^{(i)}(x_1)}{\phi^{(i)}(x_2)}$ is independent of $i$ for $i$ sufficiently large. Moreover, we require that $\phi(x_1) = \phi(x_2)$ if and only if $x_1 = x_2$.

The class of nice functions contains most of the frequently-used distribution functions. For example, normal distributions and exponential distributions are nice. The proof of this fact is included in section 2. We believe identifiability also works for Gamma distributions. But this requires a more general definition of nice functions. The result will show up in a future paper.

By identifiability we mean the following: if two sets of parameters $\Psi$, $\Psi'$ give the same model, then $\Psi$ coincides with $\Psi'$, after a permutation of the indices. We remark that here "coincide" means "coincide up to $\delta$-shift." As we discussed in Remark 1, this is sufficient for our purposes.

To be more precise, let $\mathcal{C} = \{\{\gamma_s\}_{s=1}^{S} \,|\, S \in \mathbb{Z}_{>0}, \gamma_s \in \mathbb{R}_{>0}, \sum_{s=1}^{S} \gamma_s = 1\}$. Suppose for two sequences $\{\gamma_s\}_{s=1}^{S}$ and $\{\gamma_s'\}_{s=1}^{S'}$, we have:

$$\sum_{s=1}^{S} \gamma_s \Pr(\pi | X^{(n)}, Z, \Psi) = \sum_{s=1}^{S'} \gamma_s' \Pr(\pi | X^{(n)}, Z, \Psi') \qquad (4)$$

for all possible orders $\pi$ of $M$ products, and for all agents $n$. Then we must have $S = S'$ and (up to a permutation of indices $\{1, \cdots, S\}$) $\gamma_s = \gamma_s'$ and $\Psi = \Psi'$ (up to $\delta$-shift).

Let's fix the number of agent characteristics, $K$, for a moment. One quick observation is that the number $x_n(k)$, for any characteristic $k$, reflects certain characteristic of the agent $n$. Varying the agent $n$, this amount $x_n(k)$ is in a bounded interval in $\mathbb{R}$. Suppose the collection of data $D$ is sufficiently large. Based on this, assuming that $N$ can be be arbitrarily large, we can assume that the $x_n(k)$'s form a dense subset in a closed interval $I_k \subset \mathbb{R}$.

Hence, the equation (4) should hold for any $X \in I_k$, leading to the following problem:

**Theorem 2.** *Define an $(M-1) \times L$ matrix $\widetilde{Z}$ by setting $\widetilde{Z}_{m,l} = z_m(l) - z_M(l)$. Suppose the matrix $\widetilde{Z}$ has rank $L$, and suppose,*

$$\sum_{s=1}^{S} \gamma_s \Pr(\pi | X, Z, \Psi) = \sum_{s=1}^{S'} \gamma_s' \Pr(\pi | X, Z, \Psi'), \qquad (5)$$

*for all $x(k) \in I_k$ and all possible orders $\pi$ of $M$ products. Here, the probability measure is associated to a nice cdf. Then we must have $S = S'$ and (up to a permutation of indices $\{1, \cdots, S\}$) $\gamma_s = \gamma_s'$ and $\Psi = \Psi'$ (up to $\delta$-shift).*

The proof of this theorem is provided in section 2. Here, we illustrate the idea for the simple case, with two alternatives ($m = 2$) and no agent or alternative characteristics ($K = L = 1$). Given this, each agent's preference is between alternatives 1 and 2. Equations (5) are merely a single identity. Unwrapping the definition, we obtain:

$$\sum_{s=1}^{S} \gamma_s \Pr(\epsilon_1 - \epsilon_2 > \delta_1 - \delta_2 + xW^s(z_1 - z_2)) = \sum_{s=1}^{S'} \gamma_s' \Pr(\epsilon_1 - \epsilon_2 > \delta_1' - \delta_2' + xW'^s(z_1 - z_2)). \quad (6)$$

Without loss of generality, we may assume $z_1 = 1$, $z_2 = 0$, and $\delta_2 = 0$. We may further assume that the interval $I = I_1$ contains 0. (Otherwise, we just need to shift $I$ and $\delta$ accordingly.) Given this, the problem reduces to the following lemma.

**Lemma 2.** *Let $\Phi(x)$ be a nice cdf. Suppose,*

$$\sum_{s=1}^{S} \gamma_s \Phi(\delta + xW^s) = \sum_{s=1}^{S'} \gamma_s' \Phi(\delta' + xW'^s), \qquad (7)$$

*for all $x$ in a closed interval $I$ containing 0. Then we must have $S = S'$, $\delta = \delta'$ and (up to a permutation of $\{1, \cdots, S\}$) $\gamma_s = \gamma_s$, $W^s = W'^s$.*

The proof of this lemma is available in section 2. By applying this into (6), we can show identifiablity for the simple case of $m = 2$ and $K = L = 1$.

Theorem 2 guarantees identifiability in the limit case that we observe agents with characteristics that are dense in an interval. Beyond the theoretical guarantee, we would in practice expect (6) to have a unique solution with a enough agents with different characteristics.

Note that, the lemma 2 itself is a new identifiability result for scalar observations from a set of truncated distributions.

## 2 Proofs

### 2.1 On Strict Logarithmic Concavity

The main purpose of this section is to establish a "strict" version of the logarithmic concavity results in Prékopa [4]. As an application, we shall prove Theorem 1.

Let us first prove Lemma 1. It is a direct consequence of Theorem 9 in [4]. Since its proof inspires our work on strict log-concavity, it is worth presenting here.

*Proof of Lemma 1.* Similar to approach in [4], we consider sets $H(\vec{\psi}) = \{\vec{\epsilon} \mid g_m(\psi, \vec{\epsilon}) \geq 0, \ m = 1, \cdots, M - 1\}$. Then $L(\vec{\psi}) = Pr(\vec{\epsilon} \in H(\vec{\psi}))$. We also have $H(\lambda\vec{\psi} + (1 - \lambda)\vec{\psi}') = \lambda H(\vec{\psi}) + (1 - \lambda)H(\vec{\psi}')$ because our $g_m$'s are linear functions. By Theorem 2 in [4], the probability measure $Pr$ is strictly log-concave. So we have

$$
\begin{aligned}
L(\lambda\vec{\psi} + (1 - \lambda)\vec{\psi}') &= Pr(\vec{\epsilon} \in H(\lambda\vec{\psi} + (1 - \lambda)\vec{\psi}')) \\
&= Pr(\vec{\epsilon} \in \lambda H(\vec{\psi}) + (1 - \lambda)H(\vec{\psi}')) \\
&\geq (Pr(\vec{\epsilon} \in H(\vec{\psi})))^\lambda (Pr(\vec{\epsilon} \in H(\vec{\psi}')))^{1-\lambda} \\
&= L(\vec{\psi})^\lambda L(\vec{\psi}')^{1-\lambda}
\end{aligned} \tag{8}
$$

as desired. $\qquad\square$

However, in practice, it is important to know when the equality in 8 holds. To answer this question, we need a "strict" version of log-concavity theory.

#### 2.1.1 Strictly Logarithmic Concave Measure

Mimicing the major ideas from [4], we define strictly log-concave measures and strictly log-concave functions. Roughly speaking, they are the same as log-concave measures and log-concave functions, but subject to a uniqueness condition on when the equality holds.

**Definition 2.** *A measure $P$ defined on the Borel measurable subsets of $\mathbb{R}^m$ is said to be **strictly logarithmic concave** if*

$$\Pr(\lambda A + (1 - \lambda)B) \geq \Pr(A)^\lambda \Pr(B)^{1-\lambda}$$

*for every $0 < \lambda < 1$ and for all convex subsets $A, B \subset \mathbb{R}^m$, and the equality holds if and only if $\mu(A \triangle B) = 0$. (Here $\mu$ stands for Lebesgue measure and $\triangle$ is the symmetric difference.)*

**Definition 3.** *A positive continuous function $h(x)$ on $\mathbb{R}^m$ (resp., on a convex subset $X$ of $\mathbb{R}^m$) is said to be **strictly logarithmic concave** if for every pair $x_1, x_2 \in \mathbb{R}^m$ (resp., $x_1, x_2 \in X$) and every $0 < \lambda < 1$, we have*

$$h(\lambda x_1 + (1 - \lambda)x_2) \geq h(x_1)^\lambda h(x_2)^{1-\lambda},$$

*and the equality holds if and only if $x_1 = x_2$.*

The following technical lemma is needed later.

**Lemma 3.** *(a) Let $h$ be a logarithmic concave function on $\mathbb{R}^m$. Suppose four points $x_1, x_2, y_1, y_2$ lie on the same line, with $x_1, y_1$ lie inside the line segment connecting $x_2, y_2$. Moreover assume that $\lambda x_1 + (1 - \lambda)y_1 = \lambda x_2 + (1 - \lambda)y_2$ for some $0 < \lambda < 1$. Then*

$$h(x_1)^\lambda h(y_1)^{1-\lambda} \geq h(x_2)^\lambda h(y_2)^{1-\lambda}$$

*(b) Let $h$ be a strictly logarithmic concave function on $\mathbb{R}^m$. Let $x \in \mathbb{R}^m$ and $a > 0$ be a real number. Then there exists $\epsilon > 0$ such that*

$$h(x) \geq h(y)^\lambda h(z)^{1-\lambda} + \epsilon$$

*whenever $\lambda y + (1 - \lambda)z = x$ and $d(x, z) \geq a$. Moreover, this $\epsilon$ is uniform in $x$ and $a$ if they vary in compact neighborhoods.*

*Proof.* (a) Let $\lambda_1 = \frac{y_2 - x_1}{y_2 - x_2}$ and $\lambda_2 = \frac{y_2 - y_1}{y_2 - x_2}$. Then $0 < \lambda_1, \lambda_2 < 1$ and

$$x_1 = \lambda_1 x_2 + (1 - \lambda_1)y_2,$$

$$y_1 = \lambda_2 x_2 + (1 - \lambda_2)y_2.$$

By log-concavity, we have

$$h(x_1) \geq h(x_2)^{\lambda_1} h(y_2)^{1 - \lambda_1}$$

and

$$h(y_1) \geq h(x_2)^{\lambda_2} h(y_2)^{1 - \lambda_2}$$

So

$$h(x_1)^{\lambda} h(y_1)^{1 - \lambda} \geq h(x_2)^{\lambda \lambda_1 + (1 - \lambda)\lambda_2} h(y_2)^{\lambda(1 - \lambda_1) + (1 - \lambda)(1 - \lambda_2)}$$

Part (a) follows from the fact that $\lambda\lambda_1 + (1 - \lambda)\lambda_2 = \lambda$ and $\lambda(1 - \lambda_1) + (1 - \lambda)(1 - \lambda_2) = 1 - \lambda$.

(b) If $d(x, z) \geq a$, then $h(x) > h(y)^{\lambda} h(z)^{1 - \lambda}$ due to strict log-concavity. By part (a), $h(x) - h(y)^{\lambda} h(z)^{1 - \lambda}$ is the smallest when $d(x, z) = a$. Define a function

$$g(y, z) := h(x) - h(y)^{\lambda} h(z)^{1 - \lambda}$$

It is a continuous function on $\mathbb{R}^m \times \mathbb{R}^m$ and it is positive on the compact set

$$U := \{(y, z) \in \mathbb{R}^{2m} \mid d(x, z) = a, \ \lambda y + (1 - \lambda)z = x\}$$

So it achieves a minimum $\epsilon > 0$ on $U$. This $\epsilon$ is exactly the one we desired.

Finally, the uniformity of $\epsilon$ follows from the continuity of $g$ and the fact that $U$ is contained in a ball of radius $\max\{a, (1 - \lambda)a/\lambda\}$ centered at $(x, x)$.

$\square$

Finally, we present the following generalization of Theorem 2 in [4].

**Theorem 3.** *Let $P$ be a probability measure on $\mathbb{R}^m$ generated by a probability density of the form $f(x) = e^{-Q(x)}$ where $Q(x)$ is a strictly convex function. (Namely, $f$ is a strictly logarithmic concave function.) Then $P$ is a strictly logarithmic concave probability measure.*

*Proof.* First, we recall the following result used in the proof of Theorem 2 in [4]. This is the inequality (2.4) in [4].

**Lemma 4.** *Let $f, g$ be nonnegative Borel measurable functions on $\mathbb{R}^m$ and $0 < \lambda < 1$ be a real number. Let*

$$r(t) := \sup_{\lambda x + (1 - \lambda)y = t} f(x)g(y).$$

*Then we have inequality*

$$\int_{\mathbb{R}^m} r(t)\, dt \geq \left(\int_{\mathbb{R}^m} f^{1/\lambda}(x)\, dx\right)^{\lambda} \left(\int_{\mathbb{R}^m} g^{1/(1 - \lambda)}(y)\, dy\right)^{1 - \lambda}.$$

Come back to the proof of the Theorem. We need to show that

$$\Pr(\lambda A + (1 - \lambda)B) > \Pr(A)^{\lambda} \Pr(B)^{1 - \lambda}$$

if $\mu(A \triangle B) > 0$.

Let $f_1(x) = f(x)$ if $x \in A$ and $f_1(x) = 0$ otherwise;
Let $f_2(x) = f(x)$ if $x \in B$ and $f_2(x) = 0$ otherwise;
Let $f_3(x) = f(x)$ if $x \in \lambda A + (1 - \lambda)B$ and $f_3(x) = 0$ otherwise.
Without loss of generality, let's assume that $\mu(A \backslash B) > 0$. Notice that the set $V := (\lambda A + (1 -$

$\lambda)B\backslash B$ has positive Lebesgue measure. Pick a closed $m$-dimensional ball $B_a(x_0)$ inside $V$ of small enough radius $a > 0$. We claim that there exist $\epsilon > 0$ such that

$$f_3(t) \geq \epsilon + \sup_{\lambda x + (1-\lambda)y = t} f_1(x)^\lambda f_2(x)^{1-\lambda}$$

for all $t \in B_{a/2}(x_0)$.

Indeed, by Lemma 3 (b), we know for each $t \in B_{a/2}(x_0)$,

$$f_3(t) > \epsilon_t + \sup_{\lambda x + (1-\lambda)y = t, \ d(t,y) > a/2} f_1(x)^\lambda f_2(y)^\lambda$$

for some $\epsilon_t > 0$. Moreover, this $\epsilon_t$ varies uniformly in the ball $B_{a/2}(x_0)$. So we can simply take $\epsilon = \inf_{t \in B_{a/2}(x_0)} \epsilon_t > 0$.

Finally, the following inequality concludes the proof:

$$\int_{\lambda A + (1-\lambda)B} f(x)\,dx = \int_{\mathbb{R}^m} f_3(t)\,dt$$

$$= \int_{\mathbb{R}^m} \left( f_3(t) - \sup_{\lambda x + (1-\lambda)y = t} f_1(x)^\lambda f_2(y)^{1-\lambda} \right) dt$$

$$+ \int_{\mathbb{R}^m} \sup_{\lambda x + (1-\lambda)y = t} f_1(x)^\lambda f_2(y)^{1-\lambda}\,dt$$

$$\geq \epsilon\mu(B_{a/2}(x_0)) + \left( \int_{\mathbb{R}^m} f_1(x)\,dx \right)^\lambda \left( \int_{\mathbb{R}^m} f_2(y)\,dy \right)^{1-\lambda}$$

$$> \left( \int_A f(x)\,dx \right)^\lambda \left( \int_B f(y)\,dy \right)^{1-\lambda}$$

$\square$

### 2.1.2 Proof of Theorem 1

*Proof of Theorem 1.* Based on the proof of Lemma 1, the equality holds if and only if inequality (8) is equality. By Theorem 3, we must have $\mu(H(\vec{\psi}^{(n)}) \triangle H(\vec{\psi}'^{(n)})) = 0$. But $H(\vec{\psi})$ are closed convex sets cut out by hyperplanes of the form

$$\epsilon_{n\pi(m)} - \epsilon_{n\pi(m+1)} \geq \delta_{\pi(m+1)} - \delta_{\pi(m)} + \sum_{k,l} x_n(k)(z_{\pi(m+1)}(l) - z_{\pi(m)}(l))W_{kl}^{s(n)}.$$

So $\mu(H(\vec{\psi}^{(n)}) \triangle H(\vec{\psi}'^{(n)})) = 0$ if and only if $H(\vec{\psi}^{(n)}) = H(\vec{\psi}'^{(n)})$, which happens if and only if

$$\delta_m - \delta_M + \sum_{k,l} x_n(k)(z_m(l) - z_M(l))W_{kl}^{s(n)} = (\delta_m)' - (\delta_M)' + \sum_{k,l} x_n(k)(z_m(l) - z_M(l))(W_{kl}^{s(n)})'$$

for all $n, k, l$ and $m = 1, \cdots, M-1$. Namely, the vector

$$\vec{\tau} = \left( (W_{kl}^s - (W_{kl}^s)')_{s,k,l}, (\delta_m - \delta_M - (\delta_m)' + (\delta_M)')_m \right) \in \mathbb{R}^{SKL+M}$$

is a solution of $\widetilde{A}\vec{\tau}^T = 0$. By our assumption, $\widetilde{A}$ has full rank. So $\vec{\tau} = 0$, which says

$$\begin{cases} \delta_m = (\delta_m)' + c & \text{where } c = \delta_M - (\delta_M)' \\ W_{kl}^s = (W_{kl}^s)' \end{cases}$$

This concludes the proof of Theorem 1.

$\square$

## 2.2 On Identifiability

The main purpose of this section is to prove Theorem 2. We first recall the definition of *nice functions*.

**Definition 4.** *Let $\phi(x)$ be a smooth pdf defined on $\mathbb{R}$ or $[0, \infty)$ and let $\Phi(x)$ be the associated cdf. For each $i > 0$, we write $\phi^{(i)}(x)$ for the $i$-th derivative of $\phi(x)$. Let $g_i(x) = \frac{\phi^{(i+1)}(x)}{\phi^{(i)}(x)}$. The function $\Phi$ is called **nice** if it satisfies one of the following two mutually exclusive conditions:*

(a) ***(Type 1)*** *For any two $x_1, x_2$, the sequence $\frac{g_i(x_1)}{g_i(x_2)}$ converges to some value in $\mathbb{R}$ (as $i \to \infty$) only if either*

- $x_1 = x_2$; *or*
- $x_1 = -x_2$ *and* $\frac{g_i(x_1)}{g_i(x_2)} \to -1$ *as* $i \to \infty$.

(b) ***(Type 2)*** *For all $x_1, x_2$, the ratio $\frac{g_i(x_1)}{g_i(x_2)}$ converges to $1$, as $i \to \infty$. Moreover, for any $x_1 \neq x_2$, there exists an odd positive number $m$ such that $\phi^{(m)}(x_1) \neq \phi^{(m)}(x_2)$.*

*Proof of Lemma 2.* Let $\phi(x)$ be the pdf associated to the cdf $\Phi(x)$. By assumption, $\phi$ is nice, which means $\phi(x)$ is of Type 1 or Type 2 as in the above definition.

Consider the Taylor expansion at 0. Note that the $(i + 1)$-th derivatives of $\Phi(\delta + xW^s)$ is just $(W^s)^{i+1}\phi^{(i)}(\delta + xW^s)$. So, the induced identity on the $(i + 1)$-th Taylor coefficient is

$$\sum_{s=1}^{S} \gamma_s (W^s)^{i+1} \phi^{(i)}(\delta) = \sum_{s=1}^{S'} \gamma'_s (W'^s)^{i+1} \phi^{(i)}(\delta') \tag{9}$$

Let us assume

$$|W^1| > |W^2| > \cdots > |W^S|,$$
$$|W'^1| > |W'^2| > \cdots > |W'^{S'}|,$$

and $|W^1| \geq |W'^1|$.

Dividing the $(i + 2)$-th coefficient by the $(i + 1)$-th coefficient, we obtain

$$\frac{\phi^{(i+1)}(\delta)}{\phi^{(i)}(\delta)} \cdot \frac{\sum_{s=1}^{S} \gamma_s (W^s)^{i+2}}{\sum_{s=1}^{S} \gamma_s (W^s)^{i+1}} = \frac{\phi^{(i+1)}(\delta')}{\phi^{(n)}(\delta')} \cdot \frac{\sum_{s=1}^{S'} \gamma'_s (W'^s)^{i+2}}{\sum_{s=1}^{S'} \gamma'_s (W'^s)^{i+1}}$$

Let $g_n(\delta) = \frac{\phi^{(i+1)}(\delta)}{\phi^{(i)}(\delta)}$. Then $\frac{g_i(\delta)}{g_i(\delta')} \to \frac{W'^1}{W^1} \in \mathbb{R}$ as $i \to \infty$. Now let's discuss Type 1 and Type 2 separately.

(i) **(Type 1)**
In this case, we must have $\delta = \delta'$, $W'^1 = W^1$ or, $\delta = -\delta'$, $W'^1 = -W^1$. However, if $i$ is odd, the equation (9) tells us that $\phi^{(i)}(\delta)$ and $\phi^{(i)}(\delta')$ must have the same sign. This rules out the possibility of $\delta = -\delta'$. Thus $\delta = \delta'$ and $W^1 = W'^1$. Now equation (9) becomes

$$\sum_{s=1}^{S} \gamma_s (W^s)^{i+1} = \sum_{s=1}^{S'} \gamma'_s (W'^s)^{i+1}.$$

A classical identifiability result concludes that $S = S'$, $\gamma_s = \gamma'_s$, and $W^s = W'^s$ for all $s$ (after a permutation).

(ii) **(Type 2)**
In this case, $\frac{W'^1}{W^1}$ must equal 1. Namely, $W^1 = W'^1$. Now look at equation (9). Since

$$\frac{g_i(\delta)}{g_i(\delta')} = \frac{\phi^{(i+1)}(\delta)/\phi^{(i+1)}(\delta')}{\phi^{(i)}(\delta)/\phi^{(i)}(\delta')} \to 1$$

as $i \to \infty$, we know that $\frac{\phi^{(i)}(\delta)}{\phi^{(i)}(\delta')}$ does not grow as fast as exponentially. So, again by the classical identifiability result, we know that $\gamma_1 = \gamma_1'$. Repeating this process, we know that $W^2 = W'^2$, $\gamma_2 = \gamma_2'$, and so on. Therefore, we also know $\phi^{(i)}(\delta) = \phi^{(i)}(\delta')$ for all odd $i$. However, by assumption, we must have $\delta = \delta'$.

$\square$

*Proof of Theorem 2.* Consider all possible permutations in which product 2 is more preferred to product 1. Define $\mathfrak{S}(1;2) := \{\pi \mid 1 \text{ shows after } 2 \text{ in the order } \pi\}$. Then

$$\sum_{s=1}^{S} \gamma_s \Pr(u_1 > u_2 | X, Z, \Psi) = \sum_{\pi \in \mathfrak{S}(1;2)} \sum_{s=1}^{S} \gamma_s \Pr(\pi | X, Z, \Psi)$$

So

$$\sum_{s=1}^{S} \gamma_s \Pr(u_1 > u_2 | X, Z, \Psi) = \sum_{s=1}^{S'} \gamma_s' \Pr(u_1 > u_2 | X, Z, \Psi')$$

Unwinding the definition, this means

$$\sum_{s=1}^{S} \gamma_s \Pr(\epsilon_2 > \epsilon_1 | \delta_1 - \delta_2 + \sum_{k,l} x(k) W_{kl}^s (z_1(l) - z_2(l)))$$

$$= \sum_{s=1}^{S'} \gamma_s' \Pr(\epsilon_2 > \epsilon_1 | \delta_1' - \delta_2' + \sum_{k,l} x(k) W_{kl}'^s (z_1(l) - z_2(l)))$$

Namely,

$$\sum_{s=1}^{S} \gamma_s \Phi(\delta_1 - \delta_2 + \sum_{k,l} x(k) W_{kl}^s (z_1(l) - z_2(l)))$$

$$= \sum_{s=1}^{S'} \gamma_s' \Phi(\delta_1' - \delta_2' + \sum_{k,l} x(k) W_{kl}'^s (z_1(l) - z_2(l)))$$

Again, we may assume all of the intervals $I_k$ contain 0. If we fix $x(2), \cdots, x(K)$, we can think of $x(1)$ as a variable. By the previous Lemma, we must have

- $S = S'$

- $\delta_1 - \delta_2 + \sum_{k \geq 2} W_{kl}^s (z_1(l) - z_2(l)) = \delta_1' - \delta_2' + \sum_{k \geq 2} W_{kl}'^s (z_1(l) - z_2(l))$

- after a permutation of $\{1, \cdots, S\}$, $\gamma_s = \gamma_s'$, and $\sum_l W_{1l}^s (z_1(l) - z_2(l)) = \sum_l W_{1l}'^s (z_1(l) - z_2(l))$.

Since $x(k)$'s can be arbitrary in the intervals $I_k$'s, we must have $\delta_1 - \delta_2 = \delta_1' - \delta_2'$ and

$$\sum_l W_{kl}^s (z_1(l) - z_2(l)) = \sum_l W_{kl}'^s (z_1(l) - z_2(l))$$

for all $1 \leq k \leq K$. Now we can repeat the above for any two products. In particular, we know that $\delta = \delta'$ (up to a shift), and

$$\sum_l (W_{kl}^s - W_{kl}'^s)(z_m(l) - z_M(l)) = 0$$

for all $1 \leq m \leq M - 1$. By assumption, the $M - 1$ by $L$ matrix $Z' = (z_m(l) - z_M(l))$ had rank $L$. So the above systems of equation has a unique solution. Namely, $W_{kl}^s = W_{kl}'^s$ for all $k, l, s$. $\square$

## 2.3 Examples of Nice CDFs

### 2.3.1 Normal Distributions

Let $\phi(x) = e^{-\frac{x^2}{2}}$. Write $\phi^{(i)}(x) = f_i(x)e^{-\frac{x^2}{2}}$. For example, $f_0(x) = 1$, $f_1(x) = -x$, and so on. We have the recursive relation $f_{i+1}(x) = -xf_i(x) + f'_{i-1}(x)$. In particular, we know that $f_i(x)$ is a polynomial in $\mathbb{R}[x]$ of degree $i$.

**Lemma 5.** *We have the following recursive relations.*

    *(a)* $f_{i+1}(x) = -xf_i(x) - (i-1)f_{i-1}(x)$

    *(b)* $f'_{i+1}(x) = -if_i(x)$

*Proof.* Assume the result holds for stage $i$. For stage $i+1$, we have

$$f_{i+2}(x) = -xf_{i+1}(x) + f'_{i+1}(x) = -xf_{i+1}(x) - if_i(x)$$

and

$$\begin{aligned}
f'_{i+2}(x) &= (-xf_{i+1}(x) - if_i(x))' \\
&= -f_{i+1}(x) - xf'_{i+1}(x) - if'_i(x) \\
&= -f_{i+1}(x) - ixf_i(x) - if'_i(x) \\
&= -f_{i+1}(x) - i(xf_i(x) + f'_i(x)) \\
&= -f_{i+1}(x) - if_{i+1}(x) \\
&= -(i+1)f_{i+1}(x)
\end{aligned}$$

$\square$

Define $g_i(x) = \frac{f_{i+1}(x)}{f_i(x)}$, which is, *a priori*, a rational function with real coefficients. Dividing $f_i(x)$ on both side of the relation (a) in the previous lemma, we obtain

$$g_i(x) = -x - \frac{i-1}{g_{i-1}(x)}$$

**Lemma 6.** *Given any $\delta \in \mathbb{R}$, the sequence $\{g_i(\delta)\}$ does not converge to any number in $\mathbb{R} \cup \{\pm\infty\}$, as $i \to \infty$.*

*Proof.* If $\{g_i(\delta)\}$ does converge to some $a \in \mathbb{R}$, then

$$a = \lim_{i \to \infty} g_i(\delta) = \lim_{i \to \infty}\left(-\delta - \frac{i-1}{g_{i-1}(\delta)}\right) = -\delta - \lim_{i \to \infty} \frac{i-1}{g_{i-1}(\delta)} \to \infty,$$

a contradiction.

On the other hand, if $g_i(x) \to +\infty$, then $-\delta - \frac{i-1}{g_{i-1}(\delta)} \to +\infty$. But it's less than $|\delta|$, a contradiction. Similarly, $g_i(\delta)$ cannot converge to $-\infty$. $\square$

**Lemma 7.** *Let $\delta, \delta'$ be two real numbers. Then $\frac{g_i(\delta)}{g_i(\delta')} \to c \in \mathbb{R}$ (as $i \to \infty$) if and only if either $c = 1$, $\delta = \delta'$ or $c = -1$, $\delta = -\delta'$.*

*Proof.* We have $g_i(\delta) + \delta = -\frac{i-1}{g_{i-1}(\delta)}$ and $g_i(\delta') + \delta' = -\frac{i-1}{g_{i-1}(\delta')}$. Let $c_i = \frac{g_i(\delta)}{g_i(\delta')}$. Then

$$\frac{g_i(\delta) + \delta}{g_i(\delta') + \delta'} = \frac{g_{i-1}(\delta')}{g_{i-1}(\delta)}.$$

Thus

$$c_i + \frac{\delta - c_i\delta'}{g_i(\delta') + \delta'} = \frac{1}{c_{i-1}}.$$

Taking limit, we get

$$\lim_{i \to \infty} \frac{\delta - c\delta'}{g_i(\delta') + \delta'} = \frac{1}{c} - c.$$

However, according to the lemma, $\frac{1}{g_i(\delta') + \delta'}$ does not converge to any real number. So we must have $\delta - c\delta' = 0$. This implies $\frac{1}{c} - c = 0$. Namely, $c = \pm 1$. If $c = 1$, we must have $\delta = \delta'$ and if $c = -1$, we get $\delta = -\delta'$.

On the other hand, it's easy to see that $\frac{g_i(\delta)}{g_i(\delta')} \equiv 1$ if $\delta = \delta'$, while $\frac{g_i(\delta)}{g_i(\delta')} = -1$ if $\delta = -\delta'$. This completes the proof. $\qquad\square$

### 2.3.2 Exponential Distributions

Let $\phi(x) = \lambda e^{-\lambda x}$ ($x \geq 0$). Then $\phi^{(i)}(x) = (-1)^i \lambda^{i+1} e^{-\lambda x}$ and $g_i(x) = \frac{\phi^{(i+1)}(x)}{\phi^{(i)}(x)} = -\lambda$, a constant! In particular, for any $x_1, x_2$, the ratio $\frac{g_i(x_1)}{g_i(x_2)}$ is always 1. Moreover, if $x_1 \neq x_2$, then $\frac{\phi^{(1)}(x_1)}{\phi^{(1)}(x_2)} = e^{\lambda(x_2 - x_1)} \neq 1$. Namely, $\phi^{(1)}(x_1) \neq \phi^{(1)}(x_2)$. Therefore, $\phi(x)$ is a nice pdf of type 2.

# 3 Algorithms

The algorithm are explained in more detail in this section.

---

**Algorithm 1** Gibbs Sampling of model parameters

---

set T=number of samples
set N=number of agents
set S=number of types
**for** $t = 1$ to $T$ **do**
    **for** $n = 1$ to $N$ **do**
        Select a random alternative $m'$ uniformly
        Compute its mean $\mu_{m'} = A^n(m', :)\psi B^n$
        Find rank of $m'$ in $\pi^n$ as $r'$
        Sample utility $u_{m'}^{n(t+1)} \sim truncEF(u^n(r'+1), u^n(r'-1))$
    **end for**
    **for** $s = 1$ to $S$ **do**
        Construct $U^s = [u^{n_1 T}, ..., u^{N_s T}]^T$ where $\mathbf{S}(n_i) = s$ for $n_i \in \{n_1, ..., n_{N_s}\}$
        Sample $\psi_s^{(t+1)} \sim (A^{sT} A^s)^{-1} A^{sT} [U^{(s)}]$
    **end for**
    Sample assignments $\mathbf{S}^{(t+1)}(n)$ using algorithm(2)(Algorithm (1) in the paper)
**end for**

---

**Algorithm 2** RJMCMC to update $\mathbf{S}^{(t+1)}(n)$ from $\mathbf{S}^{(t)}(n)$

---

Set $p_{-1}, p_0, p_{+1}$, Find $S$: number of distinct types in $\mathbf{S}^{(t)}(n)$
Propose move $\nu$ from $\{-1, 0, +1\}$ with probabilities $p_{-1}, p_0, p_{+1}$, respectively.
**case** $\nu = +1$**:**
    Select random type $M_s$ and agent $n \in M_s$ uniformly and Assign $n$ to module $M_{s_1}$ and remainder to $M_{s_2}$ and Draw vector $\alpha \sim \mathcal{N}(0, 1)$ and Propose $\psi_{s_1} = \psi_s - \alpha$ and $\psi_{s_2} = \psi_s + \alpha$ and Compute proposal $\{u^n, \pi^n\}^{(t+1)}$
    Accept $\mathbf{S}^{(t+1)}(M_{s_1}) = S + 1, \mathbf{S}^{(t+1)}(M_{s_2}) = s$ with $\mathrm{Pr}_{split}$ from update $S = S + 1$
**case** $\nu = -1$**:**
    Select two random types $M_{s_1}$ and $M_{s_2}$ and Merge into one type $M_s$ and Propose $\psi_s = (\psi_{s_1} + \psi_{s_1})/2$ and Compute proposed $\{u^n, \pi^n\}^{(i+1)}$
    Accept $\mathbf{S}^{(t+1)}(n) = s_1$ for $\forall n | \mathbf{S}^{(t)}(n) = s_2$ with $\mathrm{Pr}_{merge}$ update $S = S - 1$
**case** $\nu = 0$**:**
    Select two random types $M_{s_1}$ and $M_{s_2}$ and Move a random agent $n$ from $M_{s_1}$ to $M_{s_2}$ and Compute proposed $\{u^{(n)}, \pi^{(n)}\}^{(t+1)}$
    Accept $\mathbf{S}^{(t+1)}(n) = s_2$ with probability $\mathrm{Pr}_{mh}$
**end switch**

---