[Reviews · NeurIPS 2013]

Submitted by Assigned_Reviewer_2

This paper is related with the problem of demand estimation in multi-heterogeneous agents, specifically, to classify agents and estimate preferences of each agent type using agents’ ranking data of different alternatives. The problem is important since it has great practical value in studying underlying preference distributions of multiple agents. To tackle the problem, the authors introduce generalized random utility models (GRUM), provide RJMCMC algorithms for parameter estimation in GRUM and theoretically establish conditions for identifiability for the model. Experimental results on both synthetic and real dataset show the model’s effectiveness.

In general, this paper is a comprehensive and solid work. Not only does it provide a detailed algorithm for parameter estimation in the model, as well as experiments to verify it, but also it gives non-trivial theoretical analysis for the conditions of model’s identifiability. I have gone through most of the lengthy proof and to my knowledge found no bugs. Therefore even though the GRUM model has been proposed in an earlier work in UAI, which decreases the innovation of this work by some content, I think this paper deserves to be accepted.

Some minor suggestions:
1) The authors should clarify the relation of their work with the original GRUM models, including what the former works have done, and what needs to be analyzed more deeply (such as analysis of identifiability). These should be included in Related Literature.
2) To verify the effectiveness of the new model, the experiments’ scale can be more enlarged, since setting K=4 and L=3 (these two corresponds to feature number) includes too little information of agents and alternatives.
Summary: The paper studies an important problem, and the proposal is solid.

Submitted by Assigned_Reviewer_5

This paper addresses the problem of identifying the type of each agents from his/her partial preference data, in order to use this information to better estimate the underlying preferences for each type. The authors propose a Generalized RUM to model the behavior of such clustered agents. A reversible jump MCMC technique is used to estimate the latent variables, including the types of the agents. A theoretical analysis of the identifiability of the model and uni-modality of the likelihood posterior are presented.

Quality

There are three contributions of this paper. The new GRUM model, theoretical analysis, and inference algorithm. The model is a generalization of RUM model to multiple agents with types, which is new. Theoretical guarantees are interesting, but have limitations discussed below in the significance section. The inference algorithm is quite standard and the numerical analysis is not impressive, either on the simulated data or the real world datasets. Only the performance of estimating the number of clusters is addressed, while the main problem is in clustering agents. A much more relevant numerical simulation would be simulating how the number of misclustering depends on problem parameters such as number of clusters, how much the matrix W differ between clusters, missing data, etc.

Clarity

Some claims could be better explained.

On page 2, it is not clear what the authors mean by the first paragraph of section 1.1. Which aspects of the model eliminates unrealistic substitution patterns? and avoid the situation where removing the top choices result in the same alternative choice?

On page 2, it is not clear from the numerical results that "the clustering of types provides a better fit to real world data".

On page 5, in the definition of `nice' pdfs, $\phi^{(n)}(x)$ is used without proper definition, which makes the conditions difficult to understand. For instance, given \phi is a pdf, g_n should be non-negative. But g_n(x1)/g_n(x_2) converges to -1.

The definition of `nice' cdf's is not intuitive and no explanation is given as to why the model might not be identifiable if noise cdf is not `nice'.

On page 7, it is claimed that "It can be seen that GRUM with 3 types has significantly better performance than...". However, from the table, it seems like the gain is only marginal. How significant is the gain of 2~3 % in the log posterior?

Originality

This paper extends the definition of RUM model to the setting where there are multiple alternatives and multiple agents. The correlation between the agents are modelled via types that an agent belongs to. RJ-MCMC approach seems to be quite standard.

Significance

The main results on the theoretical guarantees are interesting, but the application is limited. For theorem 1, unimodality is only established when the types are known (as clearly explained in the paper). This limits the convergence of MCMC approach, and it is not clear how long one should run the MCMC in practice. This paper does not explain why the proposed problem is difficult. Why has this problem not been addresses so far, as the authors claim? Further, because there is no comparisons either in theoretical results or numerical results, it is difficult to judge how good the proposed algorithm is.
Summary: Theoretical guarantees are interesting, but has limitations. A comparison to fundamental limit or other approaches is lacking, either in theory or simulations.

Submitted by Assigned_Reviewer_6

the paper discussed random utility models with "Types". The definition of "type" in this work is the formula that
combines agent's attributes with those of a given alternative, giving rise to a perceived value. It doesn't necessarily
mean that two agents of the same "type" have the same taste, or preference profile. In that sense, this model is
quite expressive. the observations are complete rankings of the set of alternatives, as induced by the perceived valures.

Aside from defining this model, the theoretical contribution, as far as I can see, is as follows:
(1) identifiability of the model in case the types are known
(2) identifiability of the model in case of unobserved types for a certain class of cdfs governing the noise.

The algorithmic contribution is a RJMCMC heuristic for recovering the model parameters from the observations.
Experiments contain both synthetic data and data from a sushi response experiment from [26].

Strengths
---------
This model is new, as far as I know. The sushi experiments somewhat justifies it because the best fit
comes from assuming 3 "types", and not just "1". (see also my remark below).
The identifiability result (2) is intesting [note that identifiability result (1) is not very
surprising - it is basically the same as the full rank requirement in linear regression].

Weaknesses
----------
1. Although the model is original, I am not sure I see why latent "types" are better than, say, assuming
that each individual and each alternative have some more features that are latent. This is basically what you often do in
collaborative filtering. From a computational point of view this would give a non-convex optimization problem, but
then, so is the model here. It would have been nice to compare both approaches.

2. In section 1.2 you say that this paper allows for inference at finer levels of aggregation such as the individual level,
whereas the cited works (e.g. [7]) do not. In the experiments however, I don't see any attempt to showcase this
finer inference ability, and hence I conclude that you could have compared your results with those cited in section 1.2
in some way. I mean, it is very nice to know that the sushi data has best fit with 3 types, but this in no way supports
your claim on "individual level inference".



detailed comments
-----------------
last paragraph in page 1 (continuing on page 2) - Regarding the "unresolved issue" of "restrictive functional
assumptions about the distribution...". The reader feels like this work is about to resolve this issue, but
I don't see how. don't you still make assumptions about the "taste shock"?

section 3.1: first sentence is very bad

last sentence on page 4: which equality? put the equality in display math and refer to it using \ref{}

last sentence on page 5: why is a theorem a problem?

page 6: "a enough"---> "enough"
Summary: random utility model with "types" with statistical identifiability results, a proposed algorithm and experiments. model new, some theoretical novelty, experiments a bit disappointing.
Author Feedback

Author rebuttal: We thank all the reviewers for their insightful comments.
R: Reviewers’ Comment
A: Authors’ response

Reviewer 1:
R: This…model’s effectiveness.

R: In general, this paper is a comprehensive and solid work…I think this paper deserves to be accepted.

R:
1) The authors…

A: Thanks. We will add some discussion on this point.
A: At a high level – previous GRUM models have not considered latent types. In considering multiple types, a new inference procedure is required and model properties such as identifiability need to be revisited. To the best of our knowledge we are the first to study identifiability of a mixture model for partial ranking data.

R:
2) To verify…

A: The choice of K=4 and L=3 is solely to be consistent with the Sushi data (the only publicly available data set we have been able to identify for this kind of inference) provides only K=4 and L=3 non categorical features.
A: However, we have completed synthetic experiment results with larger scales for K and L and we can definitely add them.

Reviewer 2:

Quality
R:There are three contributions...Missing data,etc.

A: Thanks. The data set (Sushi data) is the only publicly available data set we have identified that has both full ranks (which we can use to simulate partial ranks) and characteristics for both users and alternatives.

A: We have tried some additional experiments focused on interpreting the detected types and so forth, however, the sushi data does not appear to have easily interpretable types.

A: Because of this, we have focused in the paper on the flexibility of the model in handling partial ranks and GRUM, and on the scalability of inference, which is an improvement over former techniques in econometrics. Moreover, we have collaboration with economists to extend and apply this work to econometric settings.

Clarity
R: On page 2…choice?

A: Thanks, we will clarify and provide a brief description. Modeling marginal utilities as a function of the characteristics of alternatives leads to agents' utilities to be correlated across alternatives with similar characteristics. And the introduced correlation avoids unrealistic substitution patterns.

R: On page 2...real world data".

A: We have applied the method to the sushi data set, and as presented in table 1, clustering with 3 types has a significantly better log posterior, which factors the effect of the growth in the size of parameters and plays a similar role to measures such as AIC or BIC.

R: On page 5…converges to -1.

A: We will clarify. $\phi^{(n)}(x)$ stands for the nth derivative of the pdf. This ratio can be negative.

R: The definition…is not`nice'.

A: Thanks, we will add a more intuitive description. Typical identifiability proofs use the tail behavior of the distributions, however, in our case we are dealing with truncated distributions and we use the Taylor expansion of the density to get a limit argument using the number of components in the expansion. “nice” distributions have Taylor expansion coefficients with specific growth as described in the definition; e.g., Normal and exponential distributions are “nice”.

R:On page7…log posterior?

A:The performance difference between three types and one type is statistically significant but small. We do not believe the Sushi data set is ideal --- the model is developed for customer preference behavior in markets with more type effects for example due to larger consumption decisions; e.g., in the car industry, different types buy totally different cars depending on their preferences in regard to factors such as the environment, size, and cost.

A: We are collaborating with economists to provide empirical results on real world problems in an extension of this paper.

Originality

Significance

R: The main results on the theoretical guarantees are interesting, but the application is limited.

A: We expect to find applications in ongoing work – with the growing amount of micro-level data on the ordinal preferences of individuals, there are opportunities for collaborations with the economics community.

R: For theorem1…practice.

A: The speed will be application dependent, but one good thing is that the method is parallelizable for larger data sets. For example, similar parallelization over agents and alternatives in Azari et al. NIPS12 can be used here as well.

R: This paper...Why has this problem not been addresses so far…?

A: The BLP model is used a lot within economics but hard to fit with current tools, and the extensions that we consider that Combine hidden types (clustering) and rank data have not been addressed by economists before as best we know. Bayesian inference and RJMCMC appears generally under-utilized in econometrics. We hope that this paper will lead to a sequence of case studies showing performance improvements over former methods.

A: Paragraphs 4 and 5 in the introduction have some details on this.

Reviewer 3:

Strengths
R: This model is new, as far as I know…

Weaknesses
R:
1. Although the model is original…approaches.

A: This is a fair comment. The main motivation for types is the interpretability of the model for practitioners (e.g. in econometrics you would like to categorize customers to some types which provide different preference behavior.)

R:
2. In section 1.2…"individual level inference".

A: We have experiments to test the individual level inference, however, we couldn’t conclude interpretable results and we think this is due to the limitations of this data. We will weaken the claim in the text.

detailed comments
R: last paragraph..."taste shock"?

A: It is correct that we continue to adopt a general random utility setting, however, our methodology allows for the noise distributions to be from a wider set of distributions, outside of the typical Type I extreme value distributional assumptions.

R: section3.1...
A: Thanks, we will make sure to fix them.